# Development of an incoherent broadband cavity-enhanced absorption spectrometer for measurements of ambient glyoxal and NO₂ in a polluted urban environment

Shuaixi Liang[12], Min Qin[1], Pinhua Xie[123], Jun Duan[1], Wu Fang[1], Yabai He[1], Jin Xu[1], Jingwei Liu[4], Xin Li[4], Ke Tang[12], Fanhao Meng[12], Kaidi Ye[12], Jianguo Liu[123], and Wenqing Liu[123]

[1] Key Laboratory of Environmental Optics and Technology, Anhui Institute of Optics and Fine Mechanics, Chinese Academy of Sciences, Hefei 230031, China
[2] University of Science and Technology of China, Hefei 230026, China
[3] CAS Center for Excellence in Regional Atmospheric Environment, Institute of Urban Environment, Chinese Academy of Sciences, Xiamen, 361021, China
[4] State Key Joint Laboratory of Environmental Simulation and Pollution Control, College of Environmental Sciences and Engineering, Peking University, Beijing, 100871, China

*Correspondence to:* Min Qin (mqin@aiofm.ac.cn); Pinhua Xie (phxie@aiofm.ac.cn)

**Abstract:** We report the development of an instrument for simultaneous fast measurements of glyoxal (CHOCHO) and NO₂ based on incoherent broadband cavity-enhanced absorption spectroscopy (IBBCEAS) in the $438 - 465$ nm wavelength region. The highly-reflective cavity mirrors were protected from contamination by N₂ purge gas. The reduction of the effective cavity length was calibrated by measuring collision-induced oxygen absorption at ~477 nm of pure oxygen gas input with and without the N₂ mirror purge gas. The detection limits of the developed system were evaluated to be 23 parts per trillion by volume (pptv, 2σ) for CHOCHO and 29 pptv (2σ) for NO₂ with a 30-s acquisition time, respectively. A potential cross-interference of NO₂ absorption on accurate CHOCHO measurements has been investigated in this study, as the absorption of NO₂ in the atmosphere could often be several hundred-fold higher than that of glyoxal, especially in contaminated areas. Due to nonlinear spectrometer dispersion, simulation spectra of NO₂ based on traditional convolution-simulation did not match the measurement spectra well enough. In this work, we applied actual NO₂ spectral profile measured by the same spectrometer as a reference spectral profile in subsequent atmospheric spectral analysis and retrieval of NO₂ and CHOCHO concentrations. This effectively reduced the spectral fitting residuals. The instrument was successfully deployed for 24 days of continuous measurements of CHOCHO and NO₂ in atmosphere in a comprehensive field campaign in Beijing in June 2017.

## 1. Introduction

Glyoxal (CHOCHO) is a typical intermediate for most volatile organic compounds (VOC) oxidations in the atmosphere. It plays an important role in quantifying VOC emissions, understanding VOC oxidation mechanisms, and further understanding the formation of O₃ and secondary organic aerosol (SOA). On a global scale, simulations show that biogenic isoprene is the largest source of glyoxal (47% of total contributions); anthropogenic acetylene also contributes significantly to glyoxal (20% of contributions) (Fu et al., 2008). The loss of glyoxal is mainly due to photolysis, OH and NO₃ oxidation reactions, wet and dry deposition, and irreversible absorption of water-soluble aerosols and clouds (Fu et al., 2008; Min et al.,2016). The ratio of glyoxal to formaldehyde, $R_{GF}$, is often used as an indicator of hydrocarbon precursor speciation in contaminated areas; observations in the field can give divergent conclusions (Vrekoussis et al., 2010; Kaiser et al., 2015; DiGangi et al., 2012). Glyoxal readily undergoes heterogeneous reactions to form SOA, but the contribution to SOA has a high uncertainty (Li et al., 2016; Washenfelder et al., 2011; Volkamer et al., 2007). Therefore, accurate quantification of glyoxal is a prerequisite for studies of the source, sink, and atmospheric chemistry of glyoxal.

Several technologies are currently used for measurements of glyoxal in the atmosphere, including chemical and spectroscopic methods. The common wet chemistry method is based on a derivatization reagent such as agent o-(2,3,4,5,6-pentafluorobenzyl) hydroxylamine (PFBHA), 2,4-dinitrophenylhydrazine (DNPH), or pentafluorophenyl hydrazine (PFPH), with subsequent analysis using liquid chromatography or mass spectrometry techniques (Temime et al., 2007; Ho et al., 2004; Munger et al., 1995; Pang et al., 2014). Some successful spectroscopic techniques for glyoxal include differential optical absorption spectroscopy (DOAS), laser-induced fluorescence (LIF), and incoherent broadband cavity enhanced absorption spectroscopy (IBBCEAS). Long-path DOAS (LP-DOAS) was used to measure the glyoxal concentration for the first time at a total atmospheric light-path of 4420 m with a detection limit of 0.1 parts per billion by volume (ppbv, $2\sigma$) in Mexico City (Volkamer et al., 2005a). In 2008, LP-DOAS was used to measure glyoxal above the rainforest and then compared with multi-axis DOAS (Max-DOAS), suggesting that local CHOCHO was confined to the first 500 m of the boundary layer (MacDonald et al., 2012). LIF can quantify both glyoxal and methylglyoxal with a detection limit of 2.9 pptv ($2\sigma$) in 5 min for glyoxal (Henry et al., 2012). IBBCEAS is an excellent method for measuring atmospheric trace gases. It features high sensitivity, small chemical interference, and simultaneous measurement of multiple components. IBBCEAS has been rapidly developed since Fiedler et al. first described it in 2003 (Fiedler et al. 2003). More recently, the technology has been successfully applied to measure a variety of trace gases (Min et al., 2016; Wang et al., 2017; Yi et al., 2016; Volkamer et al., 2015), weakly absorbed cross-sections of different trace gases (Chen et al., 2011; Kahan et al., 2012) and aerosol extinction (Washenfelder et al., 2013). Using a xenon arc lamp as a light source, Washenfelder et al. reported first measurement of glyoxal using the IBBCEAS technique in the laboratory with a detection limit of 58 pptv ($2\sigma$) within 1 min (Washenfelder et al., 2008). Later, Thalman et al. coupled CEAS hardware with a DOAS retrieval algorithm to measure glyoxal in open cavity mode with a detection limit of 19 pptv ($2\sigma$, 1 min) (Thalman et al., 2010). Coburn et al. subsequently measured the eddy covariance flux of glyoxal with LED-CE-DOAS for the first time and found that the nocturnal oxidation reaction on an ocean surface organic microlayer was a source of the oxygenated VOCs (Coburn et al., 2014). With significant improvements, Min et al. developed an aircraft IBBCEAS instrument and used it to measure tropospheric glyoxal with a detection limit of 34 pptv ($2\sigma$) within 5 s (Min et al., 2016). Table 1 compares different measurement techniques for glyoxal. Based on these technologies, Thalman et al. conducted a comprehensive instrument inter-comparison campaign for glyoxal (Thalman et al., 2015).

Spectral measurement techniques using broadband light sources, such as DOAS and IBBCEAS, can simultaneously observe a wide range of spectral bands during a single measurement. Thus, many contaminants can be measured concurrently. The overlap of the $NO_2$ and glyoxal absorption bands at 438–465 nm allows us to simultaneously measure their concentrations. However, $NO_2$ can interfere with the measurement of glyoxal, especially for high concentration of $NO_2$ (Thalman et al., 2015). This is a key factor that needs to be considered to improve data retrieval of glyoxal in China's highly polluted environment.

Here, we describe the development of an incoherent broadband cavity-enhanced absorption spectrometer for sensitive detection of CHOCHO and $NO_2$ in the atmosphere. The effective length of the optical cavity with purge-gas protected mirrors was accurately calibrated based on the collision-induced oxygen ($O_4$) absorption at 477 nm. The instrument detection limit was estimated using the Allan variance analysis. The effects of $NO_2$ on glyoxal were evaluated via spectral simulation and measurements. The results show that using the measured $NO_2$ reference spectrum can overcome $NO_2$ interference to glyoxal due to conventional convolution methods from the uneven dispersion of the grating spectrometer. We then applied the measured reference spectrum to the retrieval of glyoxal in the same wavelength band and obtained the glyoxal concentration in heavily polluted air in China. The IBBCEAS instrument was successfully deployed during the APHH-China (Air Pollution and Human Health in a Chinese Megacity) project, and we obtained the profiles of glyoxal and $NO_2$ concentrations in Beijing's summer atmosphere during the APHH-China campaign (2–26 June 2017).

**Table 1. Comparison of different techniques for measuring glyoxal**

| Analytical technique | Research Unit | Time resolution | Detection Limit (2σ) | Field Applications | Purge flows | Ref. |
|---|---|---|---|---|---|---|
| Microfluidic | University of York | 30 min | 53 pptv | No | / | Pang et al. (2014) |
| Mad LIP | University of Wisconsin-Madison | 1 min | 12 pptv | USA | / | Huisman et al., (2008) |
| LIP | University of Wisconsin-Madison | 5 min | 2.9 pptv | No | / | Henry et al., (2012) |
| LP-DOAS | Massachusetts Institute of Technology | 2-15 min | 0.1 ppbv | Mexico City | / | Volkamer et al., (2005) |
| IBBCEAS | University of Colorado | 1 min | 58 pptv | Laboratory | no | Washenfelder et al. (2008) |
| LED-CE-DOAS | University of Colorado | 1 min | 19 pptv | Laboratory | Yes | Thalman et al., (2010) |
| ACES | University of Colorado & NOAA | 5 s | 34 pptv | USA & China | no | Min et al., (2016) |
| **IBBCEAS** | **Anhui Institute of Optics and Fine Mechanics, CAS** | **30 s** | **23 pptv** | **China** | **Yes** | **This work** |

## 2. System and principle

### 2.1 Description of the IBBCEAS setup

The IBBCEAS technology is an absorption spectroscopy technique. It improves the effective path length via multiple light reflections in an optical cavity. This leads to a significant improvement of the detection sensitivity. Our design of the IBBCEAS setup consists of a light-emitting diode (LED) light source, a pair of off-axis parabolic mirrors, a pair of high-reflectivity cavity mirrors, Teflon perfluoroalkoxy polymer resin (PFA) optical cavity, optical band-pass filter, an optical fiber-coupled grating spectrometer and some other components. A schematic diagram of the instrument is shown in Fig. 1.

The light from a high-power blue LED (LZ1-04B2P5, LedEngin) with a peak wavelength of ~448 nm was coupled to the optical cavity via a 90 °off-axis parabolic mirror (Edmund Optics). The temperature of the LED was measured by a temperature sensor (PT1000) and controlled by a thermoelectric cooler (TEC) at 20 ℃ ± 0.1 ℃ to reduce the impact of temperature fluctuations on the LED. The optical cavity consisted of two 1-inch diameter mirrors (Advanced Thin Films) with 1-m radius of curvature, and the manufacturer stated that the reflectivity was greater than 99.995% at 455 nm. Multiple reflections of light between two high-reflectivity cavity mirrors increased the effective absorption path length. The light exiting the cavity passed through an optical band-pass filter (FB450-40, Thorlabs) to eliminate stray light. It was then focused onto a 1-m optical fiber (600 μm in diameter with a numerical aperture of 0.22) by a second off-axis parabolic mirror. Finally, the other end of the fiber cable was coupled to a compact Czerny-Turner spectrometer (Ocean Optics, QE65000) with a spectral resolution of ~0.57 nm around 450 nm. The CCD in the QE65000 spectrograph is thermally regulated at -10.0 °C to minimize the dark current. A 2-μm teflon polytetrafluoroethylene (PTFE) membrane filter (Tisch Scientific) was used in the front of the inlet to remove aerosols—this reduced scattering losses by particulate matter and its impacts on the effective path length (Thalman et al., 2010). Each cavity mirror was purged with a constant flow of dry nitrogen at a rate of 0.1 sL min$^{-1}$ (standard liters per minute) to block their contact with air samples inside the cavity. This ensured cleanness of the cavity mirror throughout the experiment.

The combination of a mass flow controller and a rotameter maintained a constant combined sample and purge gas flow rate of 1.2 sL min$^{-1}$, which resulted in a gas residence time of about 16 s in the optical cavity.

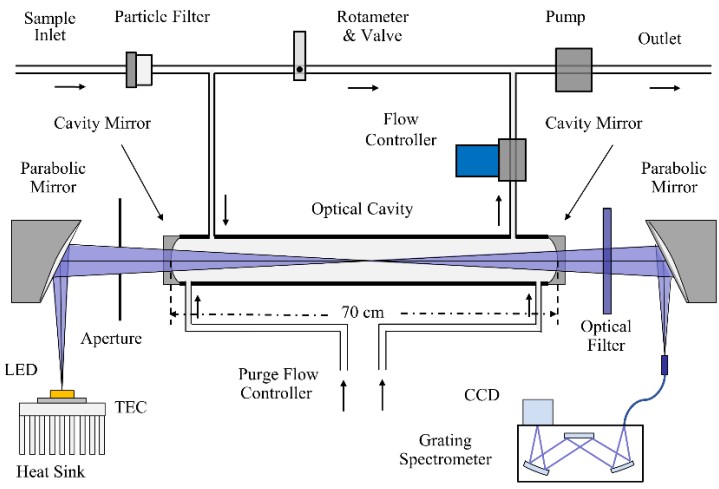

Fig. 1. Schematic of the incoherent broadband cavity enhanced absorption spectrometer.

*2.2 Theory of IBBCEAS*

The total extinction in the optical cavity includes the absorption by trace gases, Rayleigh scattering by gas molecules, and Mie scattering by particles. The use of a filter in air sampling pipeline removes the particles. The Rayleigh scattering extinction of pure $N_2$ is about $2.5 \times 10^{-7}$ cm$^{-1}$ at 455 nm, which is comparable to the cavity loss $\sim 8.1 \times 10^{-7}$ cm$^{-1}$ based on mirror reflectivity and cavity length (i.e. $(1 - R)/d$). Thus, the general description of the total optical extinction $\alpha_{abs}$ within the optical cavity is (Washenfelder et al., 2008):

$$\alpha_{abs}(\lambda) = \left( \frac{1 - R(\lambda)}{d_{eff}} + \alpha_{Ray}(\lambda) \right) \left( \frac{I_0(\lambda) - I(\lambda)}{I(\lambda)} \right), \tag{1}$$

where $R(\lambda)$ is the wavelength-dependent reflectivity of the cavity mirrors, $\alpha_{Ray}(\lambda)$ is the extinction for Rayleigh scattering, $I_0(\lambda)$ and $I(\lambda)$ are the light intensities transmitted through the optical cavity without and with the absorbing species, respectively, and $d_{eff}$ is the effective cavity length. The mirror reflectivity $R(\lambda)$ is determined from the Rayleigh scattering of $N_2$ and He via the following equation (Washenfelder et al., 2008):

$$R(\lambda) = 1 - \frac{\frac{I_{N_2}(\lambda)}{I_{He}(\lambda)} \cdot \alpha_{Ray}^{N_2}(\lambda) d_0 - \alpha_{Ray}^{He}(\lambda) d_0}{1 - \frac{I_{N_2}(\lambda)}{I_{He}(\lambda)}}, \tag{2}$$

Here, $I_{N_2}(\lambda)$ and $I_{He}(\lambda)$ are the light intensities measured when the cavity is filled with $N_2$ and He, respectively. Term $\alpha_{Ray}^{N_2}(\lambda)$ and $\alpha_{Ray}^{He}(\lambda)$ are the extinction caused by Rayleigh scatterings of $N_2$ and He, respectively. Term $d_0$ is the distance between the two cavity mirrors. Terms $d_0$ and $d_{eff}$ are not equal due to cavity mirror purging. Determination of the $d_{eff}$ will be described in the Section 3.2. After obtaining the mirror reflectivity $R(\lambda)$, the absorption coefficient $\alpha_{abs}$ can be calculated according to Eq. (1). If the chamber contains a variety of gas absorbers (including $NO_2$ and CHOCHO), then the absorption coefficient $\alpha_{abs}$ will be the sum of their individual contributions and can be written via the following equation:

$$\alpha_{abs}(\lambda) = \sum_i^n \alpha_i(\lambda) = \sum_i^n \sigma_i(\lambda)N_i = \sigma_{NO_2}(\lambda)[NO_2] + \sigma_{CHOCHO}(\lambda)[CHOCHO] + \ldots, \tag{3}$$

Here, $\sigma_i(\lambda)$ and $N_i$ are the absorption cross-section and number density for the i-th trace absorber, and n is the total number of absorbers. Finally, the absorber concentrations can be retrieved from the measured broadband spectrum via the DOASIS program (Kraus, 2006).

## 3.    Results and Analysis

### 3.1 Determination of the cavity mirror reflectivity

The cavity mirror reflectivity needs to be accurately determined for subsequent measurements of the concentrations of trace gases inside the cavity. We measure and update the value of the mirror reflectivity once every two days to ensure the reliability of the retrieval data. Using the difference of Rayleigh scattering cross-sections between $N_2$ and He, we calculated the mirror reflectivity $R(\lambda)$ according to Eq. (2). The values of $\alpha_{Ray}^{N_2}(\lambda)$ and $\alpha_{Ray}^{He}(\lambda)$ were taken from published references (Shardanand et al., 1977; Sneep et al., 2005). The black and red curves in Fig. 2 were the spectrometer's signal intensity when the cavity was filled with high purity $N_2$ (99.999%) and He (99.999%), respectively. The difference in light intensity due to Rayleigh scattering by $N_2$ versus He is well visible. The shaded spectral region (438 nm – 465 nm) indicated in the figure contains the main absorption peak of glyoxal and is of primary interest for its spectral retrieval. The mirror reflectivity at the maximum absorption position of glyoxal (455 nm) is about 0.999942. The cross-sections were obtained by convolving the high-resolution literature cross-sections of CHOCHO (Volkamer et al., 2005b), $NO_2$ (Voigt et al., 2002) and $H_2O$ with the nominal spectrometers's instrument function of 0.57 nm full width at half maximum (FWHM). The $H_2O$ absorption cross section was calculated with the SpectraPlot program based on the HITRAN2012 database (Rothman et al., 2012).

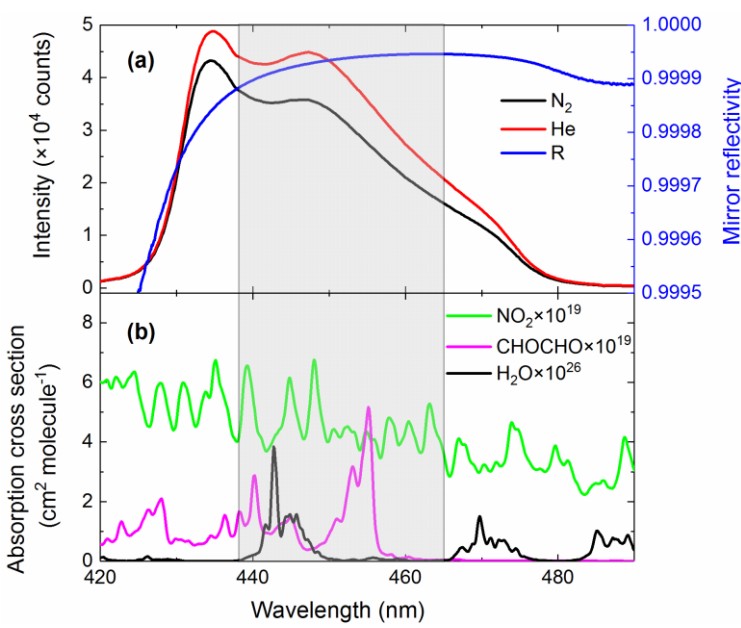

Fig. 2. (a) Calibration of the mirror reflectivity. The black and red curves represent spectrometer's CCD traces of nitrogen and helium, respectively, with a spectral acquisition time of 30 s. The blue line is the resulting mirror reflectivity curve. (b) The green, magenta and black lines are convolution-based literature absorption cross-sections of $NO_2$, glyoxal and $H_2O$ vapor, respectively.

### 3.2 Calibration of the effective cavity length

Considering the intended application's environmental conditions of high-load particulate matter and high-concentration polluting gases, we used an aerosol filter to reduce particles entering the optical cavity and purged the immediate space in front of the cavity mirrors with pure $N_2$ gas to keep the cavity mirrors clean (see Fig. 1). This purging made it difficult to accurately measure the effective cavity length. However, the effective cavity length is required for retrieving trace gas concentrations. Here, we utilized the collision-induced oxygen absorption (referred as $O_2$-$O_2$ or $O_4$ absorption) (Thalman et al., 2013) at ~477 nm within our operation wavelength region to quantify the effective cavity length. Pure $O_2$ gas was introduced into the optical cavity and the $O_2$-$O_2$ 477-nm absorptions with and without the $N_2$ mirror purges were then measured. The $O_2$ flow rate was 1 sL min$^{-1}$ and the total $N_2$ purge flow rate was 0.2 sL min$^{-1}$. Figures 3a and 3b show an example of $O_2$-$O_2$ measurement spectrum, its model fitting, and the fit residuals. Figure 3c shows the time series of equivalent $O_2$ concentrations when $N_2$ mirror purge gas was alternated between On and Off. A coarse estimation for the cavity length reduction factor was calculated to be 0.87 at room temperature and standard atmospheric pressure according to Eq. (4).

$$d_{eff\_O4\_based} = d_0 \times \frac{[\sqrt{O_4\ Signal}\ ]_{Purge\ on}}{[\sqrt{O_4\ Signal}\ ]_{Purge\ off}}, \tag{4}$$

Here, the $O_4$ signals were the retrieve concentration of $O_4$ with and without the $N_2$ purge flows, respectively. Furthermore, we modelled the reduction factor of the effective cavity length due to purge gas to include the effect of the dilution of sample gases by purge gases inside the cavity and the fact that the measured $O_4$ spectra were proportional to the product of $[O_2]\times[O_2]$ concentrations. According to the simulation results, if $N_2$ purge gases distributed evenly to both ends of the cavity and 50% of the total purge $N_2$ was involved in the dilution of $O_2$, the reduction factor for linear absorption process was 0.841, which was 3.3% less than the coarse estimation value of 0.87. An uncertainty of the purge $N_2$ participating in the $O_2$ mixture at 40% or 60% could cause a ~2% uncertainty in the cavity length reduction factor. In this experiment, $d_0 = 70.0$ cm and the calculated $d_{eff}$ was 58.9 cm.

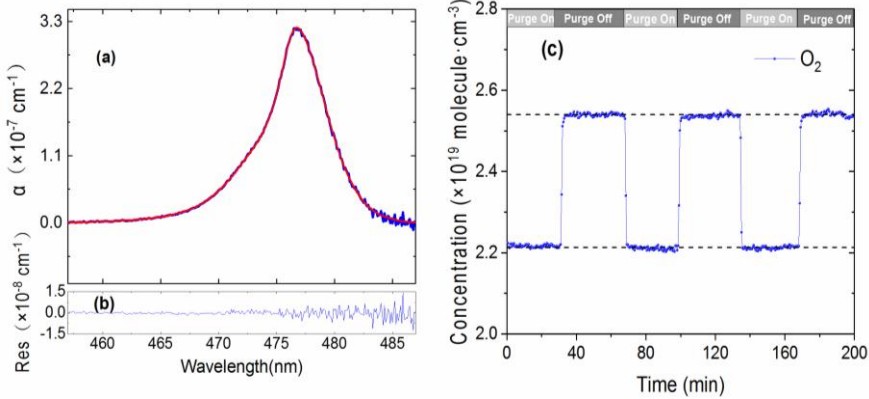

Fig. 3. (a) Example of retrieved and fitted absorption spectrum of $O_4$. The blue line is the measured spectrum and the red line is the fitted spectrum of $O_4$; (b) Fit residuals; (c) The time series of equivalent $O_2$ concentrations when $N_2$ purge gas was alternated between On and Off.

### 3.3 Instrument stability and detection limit

The stability of the system affects its detection sensitivity. An ideal stable system can theoretically achieve an extremely high sensitivity by averaging measurements over a long period of time. However, there are practical considerations that limit this to a certain time range (Werle et al., 1993). For an IBBCEAS system, its stability is mainly affected by the mechanical drifts of the system and the change in the intensity and central wavelength emission of the light source due to temperature variations.

We used two methods to describe the performance of the system: distribution analysis and Allan variance analysis of a large number of measurements. For more than 8 hours, 10,000 spectra were continuously acquired with the optical cavity filled with dry nitrogen. As the cavity was free of any $NO_2$ and CHOCHO, these measurements reveal the fluctuations around zero concentration. The acquisition time of each spectrum was 3 s (which combined 10 spectrometer's CCD traces with an exposure time of 300 ms each). The concentrations of $NO_2$ and CHOCHO time series (Fig. 4a and 4b) were obtained by retrieving the spectral measurements. The histograms (Figs. 4c and 4d) were constructed from this data (Fig. 4a and 4b) respectively. The standard deviation ($\sigma_{Gaussian}$) and mean value (μ) were calculated from the Gaussian distributions of the histograms for each gas. The mean value was an offset from the expected zero and was considered to be a residual "background". The limit of detection (LOD) can be defined as Eq. (5) from analytical chemistry and this method was also commonly used in cavity-enhanced systems to evaluate instrument performance (Thalman et al., 2015; Fang et al., 2017).

$$LOD_{exp} = 2 \times \sigma_{Gaussian} + |background| , \tag{5}$$

According to Eq. (5), the detection limits (with a 3 s acquisition time) for $NO_2$ and CHOCHO were calculated to be about 0.094 ppb (2σ) and 0.058 ppb (2σ), respectively.

Allan variance analysis has been also a convenient way to describe the stability and detection limit of a system as a function of averaging time. We used Allan variance analysis to characterize the overall stability of our system and to determine the optimum averaging time and predict the detection limit of the system. The above mentioned 10,000 spectral concentration values were divided into M groups—each containing N values (N =1, 2, …, 2000; M = 10000/N =10000/1,10000/2, …, 10000/2000). The average of N values is denoted as $y_i$ ($i$ = 1, 2, …, $M$), and the corresponding averaging time is $t_{avg} = N \times 3\ s$. Since each spectrum was measured in the optical cavity filled with dry nitrogen, the $y_i$ values contain only measurement noise as a function of averaging times (Langridge et al., 2008). The Allan variance and standard deviation of $NO_2$ and CHOCHO concentrations are calculated according to Eqs. (6) and (7), respectively, as shown in Figs. 4e and 4f. The Allan deviation initially decreases with a gradient -0.5 as averaging time increases, before it starts gradually to increase towards a longer averaging time. The optimum integration time (210 s for CHOCHO) of the instrument is around the minimum of Allan deviation. Further increase of the integration times yield no more decrease in the Allan deviation due to system drift. For a total acquisition time of 3 s, the detection limits (standard deviation) of $NO_2$ and CHOCHO are 0.083 ppbv and 0.052 ppbv (2σ), respectively. This result is consistent with $LOD_{exp}$ (0.094 ppbv and 0.058 ppbv). By increasing the spectral averaging time to 30 s (which combined 100 spectrometer's CCD traces with an exposure time of 300 ms each), the $NO_2$ and CHOCHO detection limits (standard deviation) were reduced to 29 pptv (2σ) and 23 pptv (2σ), respectively. To capture the rapid variation of CHOCHO in the field, time resolution of the IBBCEAS instrument was typically set to 30 s. During field measurements, system drift was managed by frequently measuring the $I_0$ spectrum and stabilizing the temperature of the system.

$$\sigma_A^2(t_{avg}) = \frac{1}{2(M-1)} \sum_{i=1}^{M-1} \left[ y_{i+1}(t_{avg}) - y_i(t_{avg}) \right]^2 , \tag{6}$$

$$\sigma_s^2(t_{avg}) = \frac{1}{M-1} \sum_{i=1}^{M} \left[ y_i(t_{avg}) - \mu \right]^2 , \tag{7}$$

In the above formulas, $y_i(t_{avg})$ is the averaging concentration of the $i$-th group. Term $\mu$ is the average concentration over the entire measurement period.

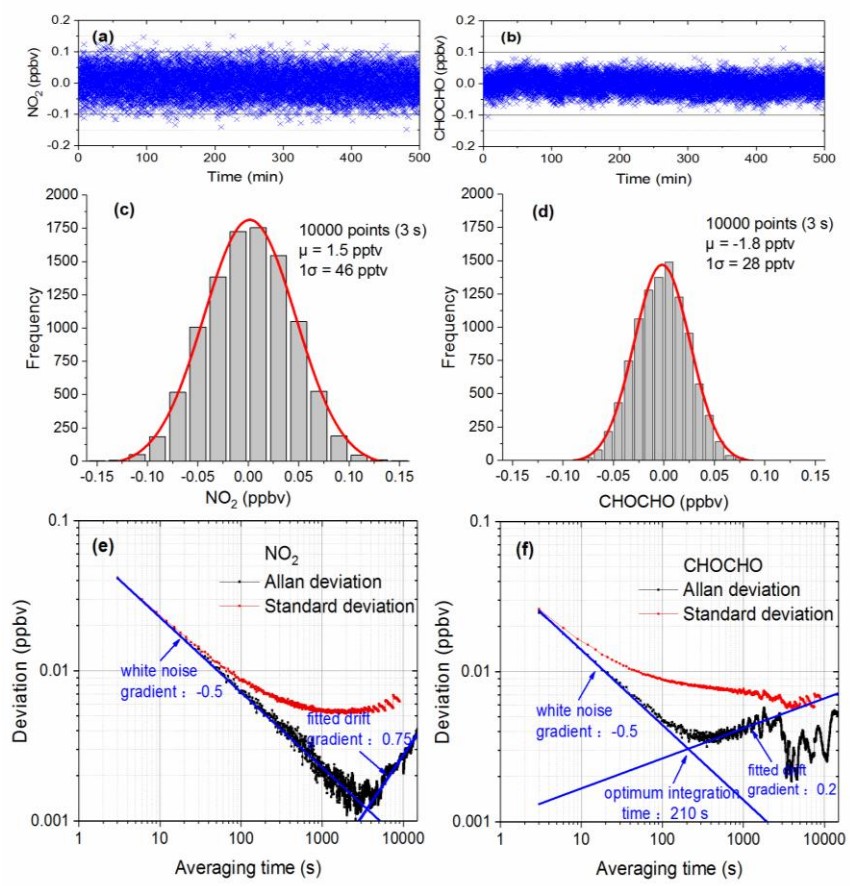

Fig. 4. Evaluation of the instrument performance. Panels (a) and (b) are the time series of NO₂ and CHOCHO with 3 s acquisition time. Panels (c) and (d) show the histogram analyses of the measurements of NO₂ and CHOCHO, respectively. Panels (e) and (f) are Allan deviation plots for measurements of NO₂ and CHOCHO, respectively.

*3.4  Sampling loss of glyoxal and Measurement of glyoxal sample gas*

In order to obtain a stable concentration of glyoxal, we used a mass flow controller to allow the quantitative high purity nitrogen through the trap containing solid glyoxal at atmospheric pressure and at -72 ℃. The sample stream out of the glyoxal trap was further diluted with dry high purity nitrogen in a sampling bag (PFA) before entering the inlet of the IBBCEAS.

*3.4.1 Sampling tube loss of glyoxal*

We measured the glyoxal sample gas in the sampling bag alternately using 3 m and 10 m sampling tubes (PFA) at a flow rate of 1 L/min to study the loss of CHOCHO in the sampling tube. The experimental results showed that sampling tube length has no obvious impact on glyoxal loss (Fig. 5). This is consistent with previous findings (Min et al., 2016).

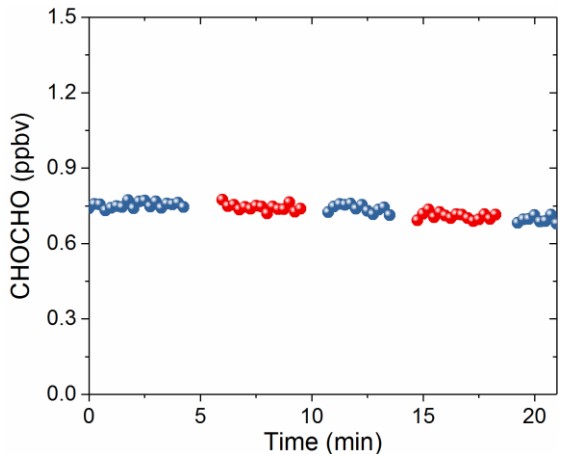

Fig. 5. Measurement of CHOCHO loss in the sampling tube. Blue dots correspond to the measured CHOCHO with the extra 3 m PFA inlet tube; Red dots correspond to the measured CHOCHO with the extra 10 m PFA inlet tube.

### 3.4.2 Measurements of CHOCHO standard additions

The high concentration of glyoxal was diluted several times in proportion to obtain the concentration time series as shown in the Fig. 6a. The last five low concentration gradients in the Fig. 6a are diluted proportionally by the first maximum concentration gradient. Figure 6b shows the average of these concentration gradients and the normalized mixing ratios, with high linearity ($R^2 = 0.9996$). Here, the normalized mixing ratio is calculated based on the dilution flows. The intercept value of -2.4 ppbv may be due to the loss of glyoxal onto the surfaces exposed to the gas samples during the experiment.

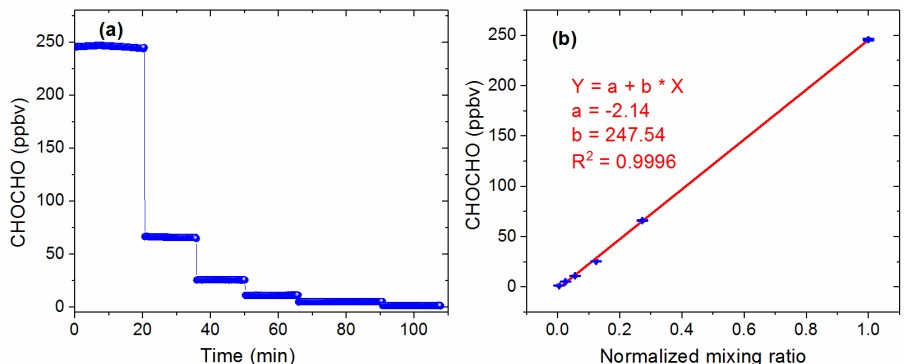

Fig. 6. Measurement of glyoxal sample gas. (a) Different concentrations of CHOCHO measured by IBBCEAS. (b) shows the correlation between the average of these concentration gradients and the normalized mixing ratio.

### 3.5 Interference from NO₂ and spectral fitting

Both the glyoxal and $NO_2$ have absorption bands in the same wavelength region as shown in Fig. 2. Therefore, it is important to select suitable absorption features for their retrieval to reduce cross interferences. Various factors, such as the performance of the instrument (e.g. the intensity wavelength range of the LED light source, the mirror reflectivity, and the spectrometer resolution), the absorption strength of the gas, the concentration level in the actual atmosphere, and the correlation between the absorption features of different gas species in the same wavelength region should be considered to obtain the best-fitting wavelength interval. Figure 7 shows the correlations matrix of absorption cross-sections of CHOCHO and $NO_2$ for a range of fitting intervals starting between 429 and 448 nm and ending between 457 and 475 nm. We hope to find an optimal fit interval with minimal correlation (Pinardi et al., 2013). In this paper, the retrieval band of glyoxal and $NO_2$ is finally 438 – 465 nm.

When the concentration of NO₂ exceeds ~12 ppbv in the actual atmosphere, the absorption due to NO₂ is more than 100-fold higher than that due to a typical 0.1 ppbv glyoxal in the atmosphere. The concentration of NO₂ in the atmosphere of polluted urban areas in China could reach tens or even hundreds of ppbv (Qin et al. 2009). Concurrently, the accumulation of NO₂ at night further challenges accurate glyoxal measurements. Therefore, accurate data analysis of the NO₂ absorption contributions became critical to reduce its impact on the determination of the glyoxal absorption and concentration. For modelling of measurement spectra, one common approach was to first determine a nominal spectrometer's resolution profile as instrumental function and then the literature reference spectrum was convoluted with the instrumental function of the spectrometer. However, we noticed that the grating spectrometer had nonuniform dispersions. We measured the wavelength dependence of the grating spectrometer's resolution by using narrow atomic emission lines of low-pressure Hg, Kr and Zn lamps. These results were summarized in the Table 2. The nonuniform dispersions make spectral modelling less accurate. Subsequently inaccurate modelling makes it difficult to overcome cross-interference of strongly-absorbed interference gases with weakly-absorbed gases of interest within the same wavelength region. A more reliable approach we used to obtain NO₂ reference spectra was to make a direct measurement of known concentrations of NO₂ standard gases with the spectrometer and further calibrate with the convolved literature reference spectrum. Samples of NO₂ in N₂ were prepared by flow dilution from a standard cylinder containing 5 ppm NO₂ in N₂. We verified the measured NO₂ reference spectrum and the convolved literature NO₂ reference spectrum by retrieval of the same NO₂ spectra. The difference was about 1.4%.

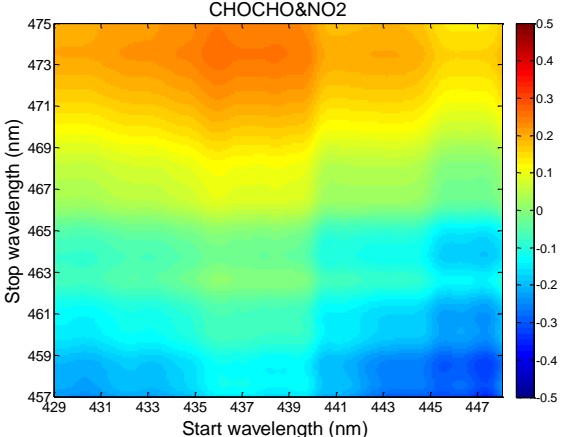

Fig. 7. Correlations matrix of absorption cross-sections of CHOCHO and NO₂ for different wavelength intervals in the 429 – 475 nm wavelength range.

**Table 2. Wavelength dependence of the grating spectrometer's resolution**

| Wavelength (nm) | 407.78 | 431.96 | 437.61 | 450.24 | 468.01 | 472.21 | 481.05 |
|---|---|---|---|---|---|---|---|
| FWHM (nm) [*] | 0.55 | 0.56 | 0.56 | 0.57 | 0.60 | 0.61 | 0.62 |

[*] The full width at half maximum (FWHM) values were determined from the emission linewidth measurements of low-pressure Hg, Kr and Zn lamps.

### 3.5.1 Residual structure from NO₂ fitting

For our application, inaccurate NO₂ fitting produces a large residual structure especially in the case of high concentrations of NO₂ (see Fig. 8). Figure 8 shows the variation characteristics of fit residuals from fitting different concentrations of standard NO₂ when using the convolution-based NO₂ reference spectrum. As is clear from Fig. 8, there is a similar residual structure in the fit residual and it increases with increasing NO₂ concentration. Such a residual structure will have a disastrous effect on

the retrieval of glyoxal in the atmosphere. Figure 9 shows that the standard deviation of these fit residuals has a good dependence with the $NO_2$ concentration.

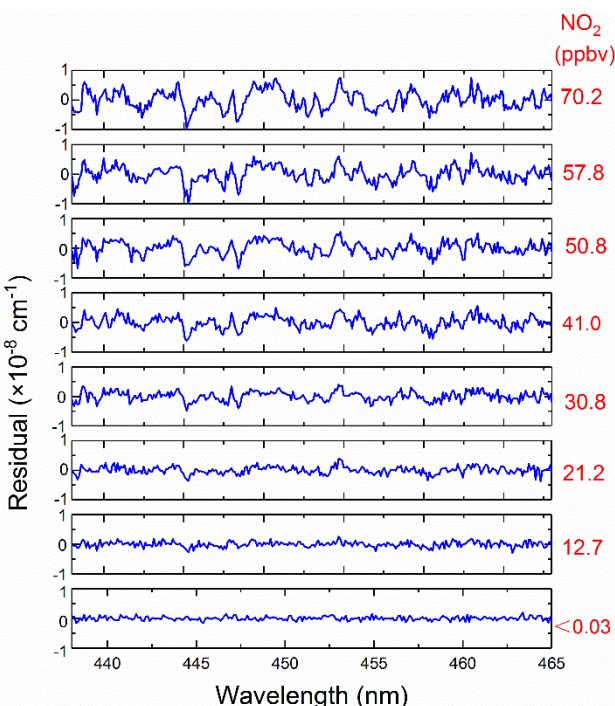

Fig. 8. Fitting residuals of different concentrations of standard $NO_2$ when using the convolution-based $NO_2$ reference spectrum

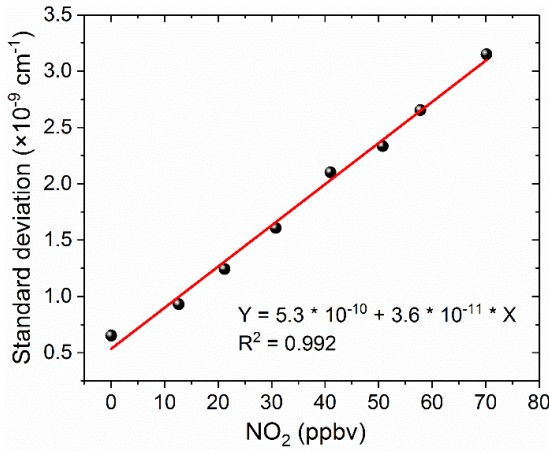

Fig. 9. The standard deviation of the fit residual from Fig.8 as a function of $NO_2$ concentrations.

### 3.5.2 Spectral simulation of $NO_2$ interference with glyoxal

The influence of residual structures in the absorption spectra has been evaluated by Stutz et al. (Stutz et al., 1996). Due to the nonuniform dispersion of the spectrometer, a stable residual structure was produced when a $NO_2$ reference spectrum based on a simple convolution calculation (by using a nominal function for the instrument line profile) was used for experimental spectral profile fitting (Fig.8). We evaluated this nonuniform dispersion effect of the co-existing $NO_2$ absorption on glyoxal spectral analysis.

The simulation spectra we used to test the accuracy of the spectral extraction comprised three components: $NO_2$ reference spectra based on measurements according to Eq. (1), as well as the Rayleigh spectrum of $N_2$ at 1atm, and convolution-simulated spectrum of 0.1 ppbv CHOCHO. We obtained simulation spectra containing different concentrations of $NO_2$ (0 − 200 ppbv)

and 0.1 ppbv glyoxal according to Eq. (3) as a summation. The spectral retrieval was conducted by using a nonlinear least-squares fitting routine. We tested the retrieval accuracy of CHOCHO by applying either a convolution-based $NO_2$ reference spectrum or a measurement-based $NO_2$ reference spectrum in the nonlinear least-squares fitting routine for the modelling of the $NO_2$ spectral contribution.

Figure 10a shows the deviation of the retrieved CHOCHO concentration from its nominal 0.1 ppbv value as a function of $NO_2$ concentration. The blue line in Fig. 10a is the retrieval result when using convolution-based $NO_2$ reference spectrum as its model function (The gray area indicates the range of fitting uncertainties). The deviation of the extracted glyoxal concentration and estimated uncertainty increase linearly as the concentration of $NO_2$ increases. The deviation of glyoxal reaches 0.58 ppbv when the concentration of $NO_2$ is 198 ppbv. In other words, for our instrument, the large bias is characterized as 2.9 pptv glyoxal/ppbv $NO_2$. Thalman et al. showed in their experiment that the CE-DOAS and BBCEAS systematic bias is 1 pptv glyoxal/ppbv $NO_2$ at higher $NO_2$ (Thalman et al., 2015). The difference between our findings and his findings may be due to differences in instruments—especially the spectrometers. When the concentration of $NO_2$ is less than ~8 ppbv, its effect on the deviation of glyoxal is less than the detection limit of the instrument (23 pptv, 2$\sigma$). The $NO_2$ likely has only a minor effect on glyoxal measurements in this low concentration case. When the retrieval is performed using the measurement-based $NO_2$ reference spectrum, the deviation of the extracted glyoxal concentration value (Fig. 10a, red line) remains close to zero. The uncertainty of the fitting error (gray area) is also small, indicating that the effect of $NO_2$ on glyoxal is negligible. Figure 10b compares the standard deviations of the fit residual as a function of $NO_2$ concentration. The standard deviation is reduced from $5.1 \times 10^{-11}$ cm$^{-1}$/ppbv $NO_2$ when using the convolution-based $NO_2$ reference spectrum in the least-squares fitting, which reduced to $1.7 \times 10^{-12}$ cm$^{-1}$/ppbv $NO_2$ when using the measurement-based $NO_2$ reference spectrum. This is an improvement of over 30 times.

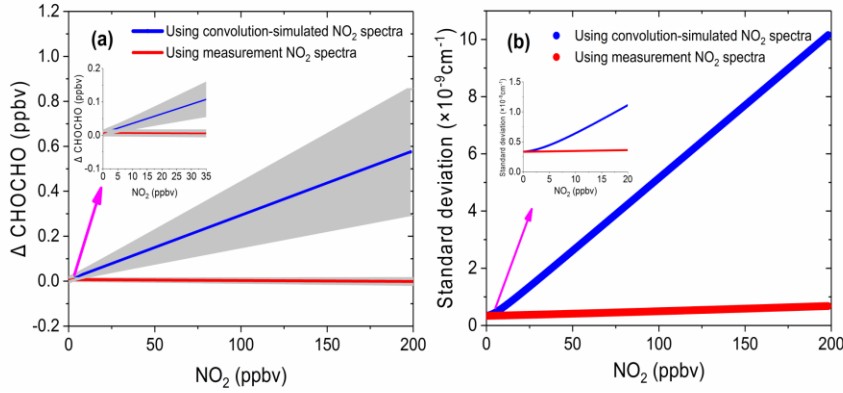

Fig. 10. The simulation results of the effect of $NO_2$ on glyoxal. (a) The deviation of the retrieved glyoxal concentration from its nominal value of 0.1 ppbv as a function of $NO_2$ concentrations. The blue line is the retrieval result using the convolution-simulated $NO_2$ reference spectra (gray area is the range of fitting uncertainty). The red line is the result of retrieval via the measured $NO_2$ reference spectrum (gray area is the range of fitting uncertainty). (b) The corresponding standard deviations of the spectral fit residual.

*3.5.3 Spectral fitting of field measurement spectra*

We compared the effects of using the convolution-based and the measurement-based $NO_2$ reference spectra to fit real atmospheric spectral measurements. As the simulation analysis in the previous Section 3.5.2 indicated already, using the measurement-based $NO_2$ reference spectrum for data analysis of the real atmospheric measurements achieved more precise $NO_2$ fitting, as both the $NO_2$ reference spectrum and the real atmospheric measurements share the same instrument (i.e. the grating spectrometer) function. The results of the real experimental spectral fitting are shown in Fig. 11. A comparison of the spectral retrievals using both the convolution-based $NO_2$ reference spectrum and the measurement-based $NO_2$ reference

spectrum are displayed in the left and right columns of Fig. 11, respectively. Corresponding fit residuals are shown in the bottom panels, and the standard deviations are 1.31 $\times 10^{-9}$ cm$^{-1}$ and 8.78 $\times 10^{-10}$ cm$^{-1}$, respectively. The standard deviation of the fitting residuals by using measurement-based NO$_2$ reference spectrum is 33% smaller than those with convolution-based reference spectrum. Moreover, the fitting residual using measurement-based NO$_2$ reference spectrum showed no obvious structure. The fitting of glyoxal is more precise, and the fitting error is reduced by 31.7% (Figs. 11g and 11h) when using the measured NO$_2$ reference spectrum. For NO$_2$, the fitting exhibits almost no difference. The result demonstrates that it is critical to use the measured NO$_2$ reference spectrum. Any tiny distortion in the NO$_2$ reference spectral profile could have severe effect on the CHOCHO extraction, because NO$_2$ absorption is about two orders of magnitude stronger than that of the CHOCHO in the local atmosphere. Figure 12 shows the standard deviation of the fitting residual of the absorption coefficient as a function of NO$_2$ concentration for measurements conducted during the APHH-China project (June 2017). The standard deviation is reduced from $5.1 \times 10^{-11}$ cm$^{-1}$/ppbv NO$_2$ to $2.2 \times 10^{-11}$ cm$^{-1}$/ppbv NO$_2$ by using the measurement-based NO$_2$ reference spectrum, which is 2.3 times smaller. The uncertainties in absorption cross-sections are 4% for NO$_2$ (Voigt et al., 2002), and 5% for CHOCHO (Volkamer et al., 2005). The difference in NO$_2$ between the literature reference spectrum and the measured reference spectrum is 1.5%. Our experimental uncertainties in cavity mirror reflectivity and effective cavity length are 5% and 2%, respectively. The propagated errors (summed in quadrature) are estimated to be 6.7% for NO$_2$ when convolution-based NO$_2$ reference spectrum was used or 6.9% when its measurement-based reference spectrum was used, and 7.3% for CHOCHO using convolution-based literature reference spectrum.

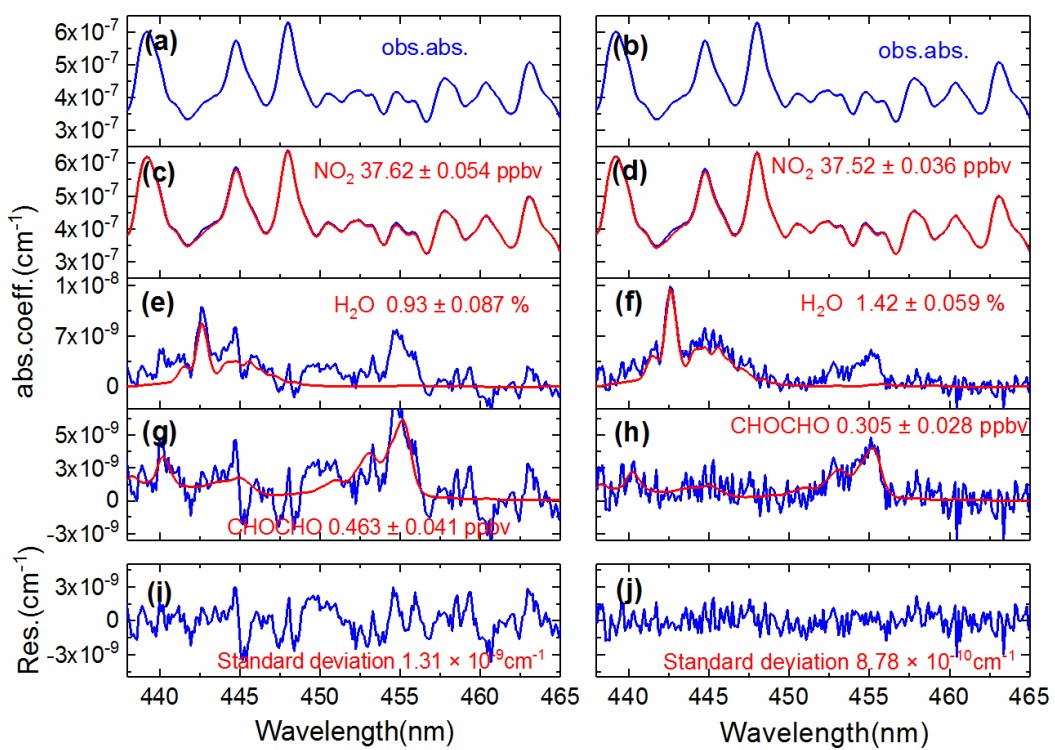

Fig. 11. A comparison of the experimental atmospheric spectral retrievals using both the convolution-based NO$_2$ reference spectrum (left column) and the measurement-based NO$_2$ reference spectrum (right column). (a) and (b) show the same atmospheric spectrum (recorded on 9 June 2017 at 12:28 local time). The retrieved NO$_2$, H$_2$O, and CHOCHO concentrations are shown in (c) and (d), (e) and (f), (g) and (h), respectively. Two overall fit residuals are shown in the bottom panels (i) and (j), with the standard deviations of 1.31 $\times 10^{-9}$ cm$^{-1}$ and 8.78 $\times 10^{-10}$ cm$^{-1}$, respectively.

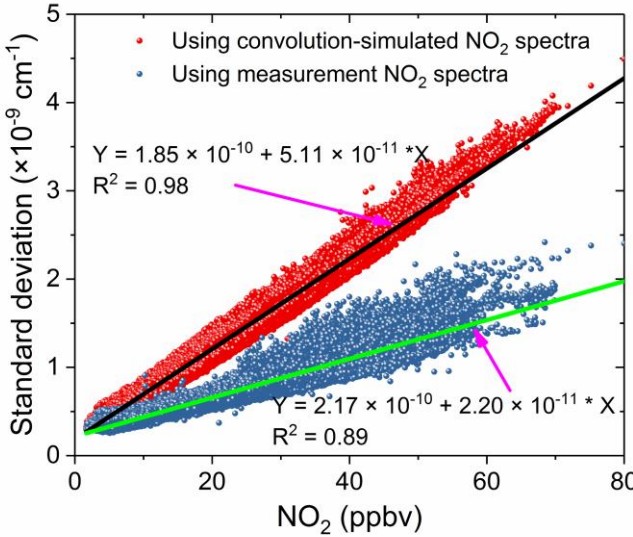

Fig. 12. The standard deviation of the fit residual of the absorption coefficients as a function of $NO_2$ concentrations for spectral data analysis during the APHH field measurements in Beijing, June 2017. The red dots were the standard deviations of the fit residuals by using a convolution-based $NO_2$ spectral profile and the black line is the linear fit of the data. The blue dots were the standard deviations of the fit residuals by using a measurement-based $NO_2$ reference spectral profile and the green line is the linear fit of the corresponding data.

*3.6 Field measurements*

The field campaign was conducted in the city Beijing at the Iron Tower Department of the Institute of Atmospheric Physics, Chinese Academy of Sciences during the APHH-China project (June 2–26, 2017). The IBBCEAS system was deployed to measure both CHOCHO and $NO_2$, supplemented by many other atmospheric measurement instruments. The sampling height of the IBBCEAS system was about 4 m above the ground. A cavity-attenuated phase shift (CAPS) spectroscopy system (University of York) for $NO_2$ data comparison was located in another container about 30 meters away from the IBBCEAS system. Figure 13 shows the 24-day continuous measurements of CHOCHO and $NO_2$ in the atmosphere by our IBBCEAS instrument. Each measurement data point was derived from each absorption spectrum acquired over 30 s (which averaged 100 spectrometer's CCD traces with an exposure time of 300 ms each). The concentration of glyoxal in the city reached 0.572 ppbv at the maximum; the average was 0.091 ppbv. Time series data for $NO_2$ measured by IBBCEAS was compared with the data from the CAPS spectroscopy system (Fig. 13b). Overall both sets of measurements were in very good agreement. The average concentration of $NO_2$ was ~20.0 ppbv and the maximum value was ~80 ppbv. A correlation plot comparing the IBBCEAS and CAPS $NO_2$ concentration data is shown in Fig. 14, with the data averaged to 1 h. The linear regression exhibits $[NO_2]$ CAPS = 1.03 $\times [NO_2]$ IBBCEAS with a correlation coefficient of $R^2 = 0.99$. Discrepancies of ~3% between the two data sets may be partly due to the different air sampling locations of these two instruments and the uncertainty of the effective cavity length calibration of the IBBCEAS. Overall this 3% deviation was within the expected 6.9% uncertainty mentioned in Section 3.5.3.

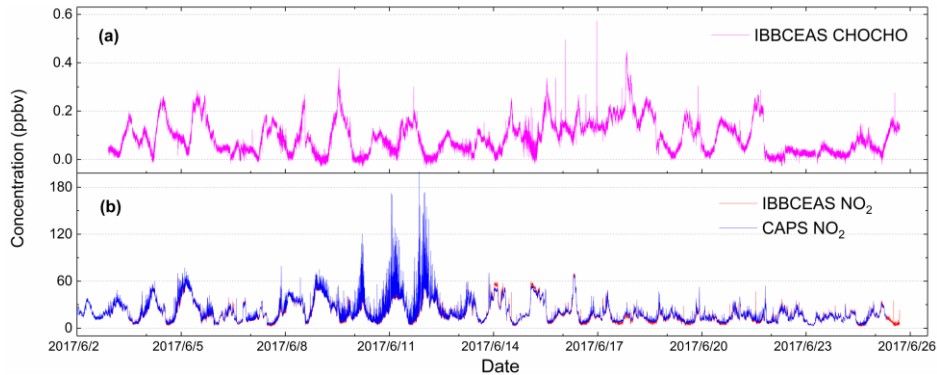

Fig. 13. Results of 24-day continuous measurements of CHOCHO and $NO_2$ in atmosphere.

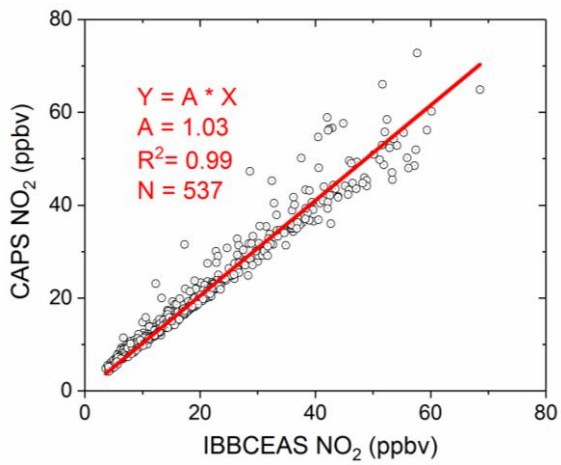

Fig. 14. Correlation plot of $NO_2$ concentration values between IBBCEAS and CAPS measurements. Each $NO_2$ data point was represented by averaged values to 1 h. The slope of the straight line fit (red line) is 1.03 with $R^2 = 0.99$

## 4. Conclusions

This paper describes the development of an IBBCEAS system and its field application to high-sensitivity measurements of atmospheric glyoxal and $NO_2$. The mirror reflectivity of the optical cavity was calibrated using the difference in Rayleigh scattering cross-sections between $N_2$ and He gases. The mirror reflectivity R is greater than 0.99994 at 455 nm, and the corresponding effective absorption pathlength is about 11.7 km (cavity dimension 0.7 m, in the absence of Rayleigh scattering). To accurately obtain a reduction factor for the cavity length when the cavity mirrors were protected by $N_2$ pure gases, the $O_4$ absorption in pure oxygen (at the 477 nm band) was used to calibrate the effective cavity length. The reduction factor of the cavity length was 0.841 at an inlet flow rate of 1 sL $min^{-1}$ and a total purge flow rate of 0.2 sL $min^{-1}$. Here, the cavity length $d_0 = 70$ cm, and the calculated $d_{eff} = 58.9$ cm. We used Allan variance analysis to identify the system's detection limits for $NO_2$ and CHOCHO. They were 0.083 ppbv ($2\sigma$) and 0.052 ppbv ($2\sigma$) at a 3-s time resolution in the laboratory, respectively. Further increases in acquisition time to 30 s improve the detection limits of CHOCHO and $NO_2$ to 23 pptv ($2\sigma$) and 29 pptv ($2\sigma$), respectively. The overall uncertainties of the instrument are 6.7% or 6.9% for $NO_2$ using convolution-based or measurement-based reference spectrum and 7.3% for CHOCHO. The effect of $NO_2$ on glyoxal was evaluated via spectral simulations and measurements. When using a convolution-based $NO_2$ reference spectral profile, the high concentration of $NO_2$ had a large effect on glyoxal and the bias was characterized as 2.9 pptv glyoxal/ppbv $NO_2$. The effect of $NO_2$ on glyoxal became negligible when retrieval was performed using the measurement-based $NO_2$ reference spectral profile. The measured $NO_2$ reference spectrum was applied to the retrieval of the actual atmospheric spectrum effectively reducing the impact of $NO_2$ on the retrieval

of CHOCHO during the APHH-China field measurement project (June 2–26, 2017). The standard deviation of the fitting residual was reduced from $5.1 \times 10^{-11}$ cm$^{-1}$/ppbv $NO_2$ to $2.2 \times 10^{-11}$ cm$^{-1}$/ppbv $NO_2$ by using the measured $NO_2$ reference spectrum, which is 2.3 times smaller. The concentrations of CHOCHO and $NO_2$ in the Beijing summer atmosphere were obtained during the APHH-China project. There was good agreement in $NO_2$ concentrations acquired by the IBBCEAS and another independent instrument using a different measurement technique–CAPS. The maximum concentrations of glyoxal and $NO_2$ in Beijing in summer reached 0.572 ppbv and ~80 ppbv respectively. This has demonstrated that our IBBCEAS instrument is capable of making accurate continuous measurements in atmospheric environments of high-load particulate matters and high-concentration polluting gases.

**Acknowledgements**

This work was supported by the National Natural Science Foundation of China (Grant No. 91544104, 41705015 and 41571130023), the Science and Technology Major Special Project of Anhui Province, China (16030801120) and the National Key R&D Program of China (2017YFC0209400). The co-author Y. He is also associated with Macquarie University, Australia. The authors would like to thank Lee James from the University of York for providing $NO_2$ data. We gratefully acknowledge the discussions with Bin Ouyang from the Lancaster Environment Centre, Lancaster University, UK.

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
