# Peer review of "Development of an incoherent broadband cavity-enhanced absorption spectrometer for measurements of ambient glyoxal and NO2 in a polluted urban environment"

_Atmospheric Measurement Techniques, 2018_

## Referee Comment (RC1) · Anonymous Referee #1 · 21 Jan 2019

Liang et al., (2018) presented an IBBCEAS at 425-475 nm for measuring the ambient NO2 and glyoxal simultaneously. The detection capacity is as good as previously works reported by Min et al., (2016). This study showed the improvement of retrieving NO2 and glyoxal by measuring and applied the NO2 cross section in field measurement, as the convolved NO2 cross section affected the retrieve of glyoxal due to the grating spectrometer had nonuniform dispersions when NO2 is high. While the paper seems missed several important details, such as the introduction of the retrieve method (DOASIS or others?); the production of the glyoxal standard, the experimental descrip-

tion of the measurement of NO2 cross section in field condition without the absorption of glyoxal. Additionally, the cross section of glyoxal is encouraged to measure by the developed IBBCEAS system and glyoxal standard at lab, which may also improve the measurement. All the details should be given and the following comments should be addressed before published at AMT.

What is the purpose of section 3.4.2, the five lower normalized CHOCHO concentrations are calculated by the dilution flow? What the offset -2.14 ppb mean in figure 6(b)?

Page 3 line 18, here the purge flow is added in each end of cavity with the same flow rate of 0.1 sL/min?

The details information of high reflectivity mirrors should be given, such as the radius of curvature of mirrors, as well as the details of the LED.

What is the role of rotameter value in this system in Figure 1?

How about the change frequency of the filter membrane in field measurement?

Eqs(3): please add $(\lambda)$ as $\alpha\_abs(\lambda)$ and $\sigma\_i\,(\lambda)$

Page 5 line 16, the HITRAN database 2012 should add the reference.

Page 5 line 11, reword "greater than 0.99994" and give the exact value.

Page 6 Eqs(4), the O4 signal is the measured spectrum signal or the retrieve concentration of O4 at purge on or off condition?

Figure 4, why the same dataset for the NO2 and glyoxal Allan variance has such a big difference?

Figure 6, is the normalized mixing ratio calculated by the dilution flows?

This paper highlights the importance of the using of measurement-based NO2 reference spectrum, while the determination of the measurement-based NO2 reference

spectrum is missed, how about the NO2 standard and the quantification of NO2 standard.

Page 13, line 13-14 this sentence is confuse, please reword it.

Page 11, line 4, Fig. 9. The standard deviation of the fit residual from fig.7, fig. 7 change to fig. 8.

---

## Referee Comment (RC2) · Anonymous Referee #2 · 24 Jan 2019

This paper describes the development of incoherent broadband cavity-enhanced absorption spectrometer (IBBCEAS) for simultaneously measuring CHOCHO and NO2 in a polluted atmosphere in extractive mode. The study and is results are very interesting especially the continuous measurements made in the city of Beijing during summer of 2017. Also of interest is the use of measured absorption cross-section of NO2 to avoid non-linear absorption effects of the CCD array detector. The manuscript is suitable for publication in AMT. The following are my specific comments, and I suggest minor revision to address these queries before publishing the manuscript.

[Figure]

1. Page-4: In the experimental setup, more details of the components may be of benefit to readers, for eg., makes and models, LED power details, cavity high-reflective mirrors' diameter, radius of curvature, manufacturer specified reflectivity at a specified wavelength, was the ccd array TE cooled and if so to what temperature, etc. Cavity (mirror-to-mirror) length may also be indicated in the schematic figure (Fig. 1)

2. In the experimental details, it may be specified whether the optical alignment was stable throughout or occasional alignments were necessary, and if so how calibrations were ensured each time.

3. Page 5, line 16: Mention of any specific/standard non-linear fitting procedures used may be beneficial. Also did the analysis take care of any spectral shifts from different cross sections (from different sources)?

4. In Fig. 3, the noise seems to be increasing from 475 nm up. Is it due to low light levels of LED in this region?

5. Page 8, line 20: How often I0 spectrum was measured?

6. On Fig.11, panel g, The CHOCHO concentration was not legible as it falls on the peak. Could this be shifted to the right or left side?

7. Page 16, line 19: "Overall this 3% deviation....". The 7.3% uncertainty in Section 3.5.3 was for glyoxal. For NO2 shouldn't it be 6.9%? The comparison here is between CAPS and IBBCEAS measurements of NO2.

8. While NO2 line shape was measured by the CCD array used for measurements to cover for the shape differences (residuals) this was not done for glyoxal. Would it matter?

9. The last sentence of the conclusions section state that measurements under high-load PM conditions are possible. Does this mean that presence of PM is OK because aerosol filter was used? Were there any quantitative measurements to characterise sampling losses against aerosol loadings in the surrounding atmosphere?

---

## Author Comment (AC1) · 11 Mar 2019

Thank you for making valuable comments on this paper. It's our pleasure to address your comments in details below.

Reviewer # 1 Comments and suggestions: Liang et al., (2018) presented an IBBCEAS at 425-475 nm for measuring the ambient NO2 and glyoxal simultaneously. The detection capacity is as good as previously works reported by Min et al., (2016). This study showed the improvement of retrieving NO2 and glyoxal by measuring and ap-

[Figure]

plied the NO2 cross section in field measurement, as the convolved NO2 cross section affected the retrieve of glyoxal due to the grating spectrometer had nonuniform dispersions when NO2 is high. While the paper seems missed several important details, such as the introduction of the retrieve method (DOASIS or others?); the production of the glyoxal standard, the experimental description of the measurement of NO2 cross section in field condition without the absorption of glyoxal. Additionally, the cross section of glyoxal is encouraged to measure by the developed IBBCEAS system and glyoxal standard at lab, which may also improve the measurement. All the details should be given and the following comments should be addressed before published at AMT.

Reply: Thanks for the constructive comments and the recommendation of a publication of the paper after making revisions to address these comments and suggestions. We use DOASIS software to retrieve data. We re-write the sentence on line 2 of page 5 as "Finally, the absorber concentrations can be retrieved from the measured broadband spectrum via the DOASIS program (Kraus, 2006).". The standard gas generator for glyoxal was designed by Prof. Xin Li and Dr. Jingwei Liu from Peking University. The test for glyoxal sample gas was done with their help. So, we add Jingwei Liu and Xin Li as co-authors. We measured the NO2 reference spectrum in the experiment and applied it to the field test. In order to make the statement clearer, we add the sentence "Samples of NO2 in N2 were prepared by flow dilution from a standard cylinder containing 5 ppm NO2 in N2." on line 12 of page 10. Since both the measured reference spectrum and the real atmospheric measurements share the same instrument (i.e. the grating spectrometer) function, the spectral fitting effect may be improved by using the measured glyoxal reference spectrum. However, the absorption due to NO2 (above 12 ppbv) is more than 100-fold higher than that due to a typical 0.1 ppbv glyoxal in the atmosphere. And it is difficult to obtain a known accurate concentration of glyoxal standard gas.
. . . . . . . . . . . . . . . . . . . . . . . . . . . . . . . . . . . . . . . . . . . . . . . . . . . . . . . . . . . . . . . . . . . . . . . . . . . . . . . . . . . . . . . . . . . . . . . . .
Comments and suggestions: What is the purpose of section 3.4.2, the five lower normalized CHOCHO concentrations are calculated by the dilution flow? What the

offset -2.14 ppb mean in figure 6(b)?

Reply: Thanks to the reviewer's comment. The section 3.4.2 indicates the linearity of the IBBCEAS instrument response. Five low normalized CHOCHO concentrations are calculated by the dilution flow ratio. In order to make the statement clearer, we add the sentence "Here, the normalized mixing ratio is calculated based on the dilution flows." on line 8 of page 9 in the revised version. When diluting high concentrations of glyoxal gas with high purity nitrogen, we use a gas pump to mix it evenly. The material of the air pump may absorb some glyoxal. We add the sentence "The intercept value of -2.4 ppbv may be due to the loss of glyoxal onto the surfaces exposed the gas samples during the experiment." on line 8 of page 9 in the revised version. We re-write the sentence on line 7 of page 9 as "Figure 6b shows the average of these concentration gradients and the normalized mixing ratios, with high linearity (R2 = 0.9996).".

. . . . . . . . . . . . . . . . . . . . . . . . . . . . . . . . . . . . . . . . . . . . . . . . . . . . . . . . . . . . . . . . . . . . . . . . . . . . . . . . . . . . .

Comments and suggestions: Page 3 line 18, here the purge flow is added in each end of cavity with the same flow rate of 0.1 sL/min?

Reply: Yes. In order to make the statement clearer, we re-write this sentence as "Each cavity mirror was purged with the constant flow of dry nitrogen at a rate of 0.1 sL min-1...".

. . . . . . . . . . . . . . . . . . . . . . . . . . . . . . . . . . . . . . . . . . . . . . . . . . . . . . . . . . . . . . . . . . . . . . . . . . . . . . . . . . . . .

Comments and suggestions: The details information of high reflectivity mirrors should be given, such as the radius of curvature of mirrors, as well as the details of the LED.

Reply: Thanks for your suggestions. Information about the radius of curvature of high reflectivity mirrors and the model of the LED has been added in the revised version.

. . . . . . . . . . . . . . . . . . . . . . . . . . . . . . . . . . . . . . . . . . . . . . . . . . . . . . . . . . . . . . . . . . . . . . . . . . . . . . . . . . . . .

Comments and suggestions: What is the role of rotameter valve in this system in Figure 1?

Reply: The rotameter is used to increase the resistance of air in the bypass gas

line. The combination of a mass flow controller and a rotameter maintains a constant gas flow through the optical cavity. We re-write the sentence on line 20 of page 3 as "The combination of a mass flow controller and a rotameter maintained a constant combined sample and purge gas flow rate of 1.2 sL min-1...".

..................................................................................................

Comments and suggestions: How about the change frequency of the filter membrane in field measurement?

Reply: In the field test, we changed the filter membrane approximately once a day. In heavy polluted weather conditions, we will increase the frequency of replacing the filter membrane approximately twice a day.

..................................................................................................

Comments and suggestions: Eqs(3): please add $(\lambda)$ as $\_abs(\lambda)$ and $\_i(\lambda)$

Reply: Thanks for your reminding. The corresponding changes have done in the revised version.

..................................................................................................

Comments and suggestions: Page 5 line 16, the HITRAN database 2012 should add the reference.

Reply: A reference has been added in the revised manuscript.

..................................................................................................

Comments and suggestions: Page 5 line 11, reword "greater than 0.99994" and give the exact value.

Reply: An exact value is used in the revised manuscript.

..................................................................................................

Comments and suggestions: Page 6 Eqs(4), the O4 signal is the measured spectrum signal or the retrieve concentration of O4 at purge on or off condition?

Reply: The O4 signals is the retrieve concentration with and without the N2 purge flows, respectively. In order to make the statement clearer, we re-

write the sentence on line 8 of page 6 as "Here, the O4 signals were the re-trieve concentration of O4 with and without the N2 purge flows, respectively.".
………………………………………………………………………………………

Comments and suggestions: Figure 4, why the same dataset for the NO2 and glyoxal Allan variance has such a big difference?

Reply: I think the difference in the absorption cross sections between NO2 and glyoxal leads to a big difference in their Allan variances. The absorption cross section of NO2 has more and larger absorption structures in the blue light band. Therefore, NO2 is more advantageous when NO2 and glyoxal together fit the same nitrogen spectrum. As shown in Figures 4(a) and (b), the value of NO2 obtained by retrieval is larger than that of glyoxal. Since the fitted value of glyoxal is smaller than NO2, glyoxal is more susceptible than NO2 under the same external interference. So, there is a big difference in the Allan variance between NO2 and glyoxal, and the optimum integration time of the instrument for glyoxal is shorter.
………………………………………………………………………………………

Comments and suggestions: Figure 6, is the normalized mixing ratio calculated by the dilution flows?

Reply: Yes. In order to make the statement clearer, we add the sentence "Here, the normalized mixing ratio is calculated based on the dilution flows." on line 8 of page 9.
………………………………………………………………………………………

Comments and suggestions: This paper highlights the importance of the using of measurement-based NO2 reference spectrum, while the determination of the measurement-based NO2 reference spectrum is missed, how about the NO2 standard and the quantification of NO2 standard.

Reply: Thanks for the comments. In order to make the statement clearer, we add the sentence "Samples of NO2 in N2 were prepared by flow dilution from a standard cylinder containing 5 ppm NO2 in N2." on line 12 of page 10.
………………………………………………………………………………………

Comments and suggestions: Page 13, line 13-14 this sentence is confused, please reword it.

Reply: Thanks for the comments. We can use either the convolution-based $NO_2$ reference spectrum or the measured $NO_2$ reference spectrum to retrieve $NO_2$ concentration, so there are two uncertainties for $NO_2$, respectively. In order to make the statement clearer, we re-write the sentence on line 13-14 of page 13 as "The propagated errors (summed in quadrature) are estimated to be 6.7% for $NO_2$ when convolution-based $NO_2$ reference spectrum was used or 6.9% when measurement-based reference spectrum was used, and 7.3% for CHOCHO using convolution-based literature reference spectrum".

............................................................................................................

Comments and suggestions: Page 11, line 4, Fig. 9. The standard deviation of the fit residual from fig.7, fig. 7 change to fig. 8.

Reply: Thanks for your reminding. The corresponding change has done in the revised version.
* * *

---

## Author Response (AR2)

*Response to the Reviewers' comments on the manuscript:*
**Development of an incoherent broadband cavity-enhanced absorption spectrometer for measurements of ambient glyoxal and NO₂ in a polluted urban environment**

mqin@aiofm.ac.cn

Thank you for making valuable comments on this paper. It's our pleasure to address your comments in details below.

**Reviewer # 1**

**Comments and suggestions:** Liang et al., (2018) presented an IBBCEAS at 425-475 nm for measuring the ambient NO₂ and glyoxal simultaneously. The detection capacity is as good as previously works reported by Min et al., (2016). This study showed the improvement of retrieving NO₂ and glyoxal by measuring and applied the NO₂ cross section in field measurement, as the convolved NO₂ cross section affected the retrieve of glyoxal due to the grating spectrometer had nonuniform dispersions when NO₂ is high. While the paper seems missed several important details, such as the introduction of the retrieve method (DOASIS or others?); the production of the glyoxal standard, the experimental description of the measurement of NO₂ cross section in field condition without the absorption of glyoxal. Additionally, the cross section of glyoxal is encouraged to measure by the developed IBBCEAS system and glyoxal standard at lab, which may also improve the measurement. All the details should be given and the following comments should be addressed before published at AMT.

**Reply:** Thanks for the constructive comments and the recommendation of a publication of the paper after making revisions to address these comments and suggestions. We use DOASIS software to retrieve data. We re-write the sentence on line 2 of page 5 as "Finally, the absorber concentrations can be retrieved from the measured broadband spectrum via the DOASIS program (Kraus, 2006).". The standard gas generator for glyoxal was designed by Prof. Xin Li and Dr. Jingwei Liu from Peking University. The test for glyoxal sample gas was done with their help. So, we add Jingwei Liu and Xin Li as co-authors. We measured the NO₂ reference spectrum in the experiment and applied it to the field test. In order to make the statement clearer, we add the sentence "Samples of NO₂ in N₂ were prepared by flow dilution from a standard cylinder containing 5 ppm NO₂ in N₂." on line 12 of page 10. Since both the measured reference spectrum and the real atmospheric measurements share the same instrument (i.e. the grating spectrometer) function, the spectral fitting effect may be improved by using the measured glyoxal reference spectrum. However, the absorption due to NO₂ (above 12 ppbv) is more than 100-fold higher than that due to a typical 0.1 ppbv glyoxal in the atmosphere. And it is difficult to obtain a known accurate concentration of glyoxal standard gas.

..................................................................................

**Comments and suggestions:** What is the purpose of section 3.4.2, the five lower normalized CHOCHO concentrations are calculated by the dilution flow? What the offset -2.14 ppb mean in figure 6(b)?

**Reply:** Thanks to the reviewer's comment. The section 3.4.2 indicates the linearity of the IBBCEAS instrument response. Five low normalized CHOCHO concentrations are calculated by the dilution flow ratio. In order to make the statement clearer, we add the sentence "Here, the normalized mixing ratio is calculated based on the dilution flows." on line 8 of page 9 in the revised version. When diluting high concentrations of glyoxal gas with high purity nitrogen, we use a gas

pump to mix it evenly. The material of the air pump may absorb some glyoxal. We add the sentence "The intercept value of -2.4 ppbv may be due to the loss of glyoxal onto the surfaces exposed the gas samples during the experiment." on line 8 of page 9 in the revised version. We re-write the sentence on line 7 of page 9 as "Figure 6b shows the average of these concentration gradients and the normalized mixing ratios, with high linearity ($R^2 = 0.9996$).".

..................................................................................

**Comments and suggestions:** Page 3 line 18, here the purge flow is added in each end of cavity with the same flow rate of 0.1 sL/min?

**Reply:** Yes. In order to make the statement clearer, we re-write this sentence as "Each cavity mirror was purged with the constant flow of dry nitrogen at a rate of 0.1 sL min$^{-1}$…".

..................................................................................

**Comments and suggestions:** The details information of high reflectivity mirrors should be given, such as the radius of curvature of mirrors, as well as the details of the LED.

**Reply:** Thanks for your suggestions. Information about the radius of curvature of high reflectivity mirrors and the model of the LED has been added in the revised version.

..................................................................................

**Comments and suggestions:** What is the role of rotameter valve in this system in Figure 1?

**Reply:** The rotameter is used to increase the resistance of air in the bypass gas line. The combination of a mass flow controller and a rotameter maintains a constant gas flow through the optical cavity. We re-write the sentence on line 20 of page 3 as "The combination of a mass flow controller and a rotameter maintained a constant combined sample and purge gas flow rate of 1.2 sL min$^{-1}$…".

..................................................................................

**Comments and suggestions:** How about the change frequency of the filter membrane in field measurement?

**Reply:** In the field test, we changed the filter membrane approximately once a day. In heavy polluted weather conditions, we will increase the frequency of replacing the filter membrane approximately twice a day.

..................................................................................

**Comments and suggestions:** Eqs(3): please add ( $\lambda$ ) as _abs( $\lambda$ ) and _i ( $\lambda$ )

**Reply:** Thanks for your reminding. The corresponding changes have done in the revised version.

..................................................................................

**Comments and suggestions:** Page 5 line 16, the HITRAN database 2012 should add the reference.

**Reply:** A reference has been added in the revised manuscript.

..................................................................................

**Comments and suggestions:** Page 5 line 11, reword "greater than 0.99994" and give the exact value.

**Reply:** An exact value is used in the revised manuscript.

..................................................................................

**Comments and suggestions:** Page 6 Eqs(4), the $O_4$ signal is the measured spectrum signal or the retrieve concentration of $O_4$ at purge on or off condition?

**Reply:** The $O_4$ signals is the retrieve concentration with and without the $N_2$ purge flows, respectively. In order to make the statement clearer, we re-write the sentence on line 8 of page 6 as "Here, the $O_4$ signals were the retrieve concentration of $O_4$ with and without the $N_2$ purge flows, respectively.".

....................................................................................

**Comments and suggestions:** Figure 4, why the same dataset for the $NO_2$ and glyoxal Allan variance has such a big difference?

**Reply:** I think the difference in the absorption cross sections between $NO_2$ and glyoxal leads to a big difference in their Allan variances. The absorption cross section of $NO_2$ has more and larger absorption structures in the blue light band. Therefore, $NO_2$ is more advantageous when $NO_2$ and glyoxal together fit the same nitrogen spectrum. As shown in Figures 4(a) and (b), the value of $NO_2$ obtained by retrieval is larger than that of glyoxal. Since the fitted value of glyoxal is smaller than $NO_2$, glyoxal is more susceptible than $NO_2$ under the same external interference. So, there is a big difference in the Allan variance between $NO_2$ and glyoxal, and the optimum integration time of the instrument for glyoxal is shorter.

....................................................................................

**Comments and suggestions:** Figure 6, is the normalized mixing ratio calculated by the dilution flows?

**Reply:** Yes. In order to make the statement clearer, we add the sentence "Here, the normalized mixing ratio is calculated based on the dilution flows." on line 8 of page 9.

....................................................................................

**Comments and suggestions:** This paper highlights the importance of the using of measurement-based $NO_2$ reference spectrum, while the determination of the measurement-based $NO_2$ reference spectrum is missed, how about the $NO_2$ standard and the quantification of $NO_2$ standard.

**Reply:** Thanks for the comments. In order to make the statement clearer, we add the sentence "Samples of $NO_2$ in $N_2$ were prepared by flow dilution from a standard cylinder containing 5 ppm $NO_2$ in $N_2$." on line 12 of page 10.

....................................................................................

**Comments and suggestions:** Page 13, line 13-14 this sentence is confused, please reword it.

**Reply:** Thanks for the comments. We can use either the convolution-based $NO_2$ reference spectrum or the measured $NO_2$ reference spectrum to retrieve $NO_2$ concentration, so there are two uncertainties for $NO_2$, respectively. In order to make the statement clearer, we re-write the sentence on line 13-14 of page 13 as "The propagated errors (summed in quadrature) are estimated to be 6.7% for $NO_2$ when convolution-based $NO_2$ reference spectrum was used or 6.9% when measurement-based reference spectrum was used, and 7.3% for CHOCHO using convolution-based literature reference spectrum".

....................................................................................

**Comments and suggestions:** Page 11, line 4, Fig. 9. The standard deviation of the fit residual from fig.7, fig. 7 change to fig. 8.

**Reply:** Thanks for your reminding. The corresponding change has done in the revised version.
* * *
**Reviewer # 2**

**Comments and suggestions:** This paper describes the development of incoherent broadband cavity-enhanced absorption spectrometer (IBBCEAS) for simultaneously measuring CHOCHO and $NO_2$ in a polluted atmosphere in extractive mode. The study and is results are very interesting especially the continuous measurements made in the city of Beijing during summer of 2017. Also of interest is the use of measured absorption cross-section of NO2 to avoid non-linear absorption effects of the CCD array detector. The manuscript is suitable for publication in AMT. The following are my specific comments, and I suggest minor revision to address these queries before publishing the manuscript.

**Reply:** Thanks for recommending a publication of the paper with minor revisions.
……………………………………………………………………………

**Comments and suggestions:** 1. Page-4: In the experimental setup, more details of the components may be of benefit to readers, for eg., makes and models, LED power details, cavity high-reflective mirrors' diameter, radius of curvature, manufacturer specified reflectivity at a specified wavelength, was the ccd array TE cooled and if so to what temperature, etc. Cavity (mirror-to-mirror) length may also be indicated in the schematic figure (Fig. 1)

**Reply:** Thanks for your suggestions. These details will be described in the revised manuscript.
……………………………………………………………………………

**Comments and suggestions:** 2. In the experimental details, it may be specified whether the optical alignment was stable throughout or occasional alignments were necessary, and if so how calibrations were ensured each time.

**Reply:** Thanks to the reviewer's comment. In my opinion, the change in the mirror reflectivity can reflect the situation of optical alignment. We have added a sentence on line 7 of page 5 as "We measure and update the value of mirror reflectivity once every two days to ensure the reliability of the retrieval data".
……………………………………………………………………………

**Comments and suggestions:** 3. Page 5, line 16: Mention of any specific/standard non-linear fitting procedures used may be beneficial. Also did the analysis take care of any spectral shifts from different cross sections (from different sources)?

**Reply:** Thanks for your reminding. we re-write the sentence on line 2 of page 5 as "Finally, the absorber concentrations can be retrieved from the measured broadband spectrum via the DOASIS program (Kraus, 2006).". The change in temperature has an effect on the gas absorption cross section. In the field test, we stabilize the indoor temperature at about 20 °C to reduce the change of the absorption cross section caused by the temperature change.
……………………………………………………………………………

**Comments and suggestions:** 4. In Fig. 3, the noise seems to be increasing from 475 nm up. Is it due to low light levels of LED in this region?

**Reply:** Yes. It can be seen from the spectrometer's CCD trace of nitrogen or helium in figure 2 that the light intensity is already low in the range above 475 nm.
……………………………………………………………………………

**Comments and suggestions:** 5. Page 8, line 20: How often $I_0$ spectrum was measured?

**Reply:** Thanks to the reviewer's comment. By adjusting three mass flow controllers, we achieved measurement and replaced the $I_0$ spectrum once an hour.
……………………………………………………………………………

**Comments and suggestions:** 6. On Fig.11, panel g, The CHOCHO concentration was not legible as it falls on the peak. Could this be shifted to the right or left side?

**Reply:** Thanks for your reminding. The corresponding change has been done in the revised version.

..............................................................................................

**Comments and suggestions:** 7. Page 16, line 19: "Overall this 3% deviation: : :.". The 7.3% uncertainty in Section 3.5.3 was for glyoxal. For NO2 shouldn't it be 6.9%? The comparison here is between CAPS and IBBCEAS measurements of NO2

**Reply:** Thanks to the reviewer's comment. It should be 6.9% here. The corresponding change has been done in the revised version.

..............................................................................................

**Comments and suggestions:** 8. While $NO_2$ line shape was measured by the CCD array used for measurements to cover for the shape differences (residuals) this was not done for glyoxal. Would it matter?

**Reply:** Thanks to the reviewer's comment. Since both the measured reference spectrum and the real atmospheric measurements share the same instrument (i.e. the grating spectrometer) function, the spectral fitting effect may be improved by using the measured glyoxal reference spectrum. However, the absorption due to $NO_2$ (above 12 ppbv) is more than 100-fold higher than that due to a typical 0.1 ppbv glyoxal in the atmosphere. And it is difficult to obtain a known accurate concentration of glyoxal standard gas.

..............................................................................................

**Comments and suggestions:** 9. The last sentence of the conclusions section state that measurements under high load PM conditions are possible. Does this mean that presence of PM is OK because aerosol filter was used? Were there any quantitative measurements to characterize sampling losses against aerosol loadings in the surrounding atmosphere?

**Reply:** Thanks to the reviewer's comment. In the use of IBBCEAS technology, it is common to use the aerosol filter membrane to remove particulate matter from the sampled air, especially under high load PM conditions. Tests in the literature have demonstrated that glyoxal has negligible losses on Teflon surface and dirty filter membrane (K.-E. Min et al., 2016; Jingwei Liu et al., 2019). In the field test, we changed the filter membrane approximately once a day. In heavy polluted weather conditions, we will increase the frequency of replacing the filter membrane approximately twice a day.
* * *
**The list of all relevant changes made in the manuscript**

1, P1. Line 4 and Line 11-12.

2, P3. Line 9, Line 12-13, Line 18, and Line 21.

3, P4. Line 1 and Line 3.

4, P5. Line 1, Line 3-4, Line 8-9, Line 15, and Line 18.

5, P6. Line 13.

6, P9. Line 7-9.

7, P10. Line 14-15.

8, P11. Line 6.

9, P13. Line 15-18.

10, P14. Line 22.

11, P16. Line 11-12, Line 43.

12, P17. Line 21-26.

[revised manuscript text omitted]